# A 1 × 2 Two-Dimensional Slanted Grating Based on Double-Layer Cylindrical Structure

**Yuda Chen, Jin Wang \*, Yihan Wang, Xingxue Li, Ge Jin and Changhe Zhou**

Institute of Photonics Technology, Jinan University, Guangzhou 510000, China
* Correspondence: wangjin@jnu.edu.cn

**Abstract:** Diffraction gratings play an increasingly important role in various planar optical systems, such as near-eye display systems for virtual reality (VR) and augmented reality (AR). The slanted gratings have more advantages than other elements. A 1 × 2 transmission two-dimensional (2D) slanted grating based on a double-layer cylindrical structure was proposed in this paper. In the initial phase of this study, this kind of grating was proposed and designed. We used rigorous coupled-wave analysis (RCWA) and simulated annealing algorithm (SA) to optimize the grating parameters. The effects of the grating geometric parameters on the diffraction efficiency were investigated using rigorous coupled-wave analysis (RCWA). The simulated annealing algorithm (SA) optimization results show that the diffraction efficiency of the (0, −1) and (−1, 0) order exceed 35% under normal incidence in the range of 429–468 nm wavelength for TE and TM polarization. Meanwhile, the total diffraction efficiency can reach up to 78%. In the last section, we discuss the tolerances for the grating parameters to ensure high quality manufacturing processes. The total effective efficiency is greater than 75% when the $MgF_2$ thickness is from 300 nm to 350 nm and the $SiO_2$ thickness is from 525 nm to 550 nm. Moreover, the grating period has a 53 nm fabrication tolerance, and the slanted angle has a 8.8-degree fabrication tolerance. The relatively large tolerances ensure that it is easy to fabricate the two-dimensional slanted grating and to achieve the targeted objectives. The proposed 2D slanted grating can be applied to 2D exit pupil expansion, which is of great importance in AR/VR applications.

**Keywords:** 2D slanted grating; double-layer cylindrical structure; high efficiency

## 1. Introduction

Diffraction gratings are the key elements in the optical devices and systems, such as spectrometers [1,2], metrology scales [3,4], wavelength division multiplexing, optical communication systems [5,6], etc. In addition, diffraction gratings play an increasingly important role in various planar optical systems, such as near-eye display systems for virtual reality (VR) and augmented reality (AR) [7,8]. The VR glasses that were originally invented used traditional optical elements. Free-form surfaces have achieved relative optimization in the aspects of evaluation dimensions of display effect, mass production, cost performance, and processing technology. However, due to the large volume of traditional optical elements and the bulky products, AR optical elements are evolving into optical waveguides. With the rapid development of micro–nano processing technology, diffractive optical waveguides have gradually attracted attention, and the products have gradually begun to be developed. Furthermore, 3D display technology is developing rapidly [9], and optical gratings are widely used in 3D display technology. Therefore, most mainstream AR manufacturers are choosing to explore innovative optical waveguide solutions to greatly reduce the size of glasses.

Many authors studied the gratings for virtual reality (VR) and augmented reality (AR) applications in display technologies. Cui et al. designed a waveguide structure with three polarizer gratings as the coupling elements to achieve large outgoing exit pupil

size [10]. Chen et al. proposed a blazed grating structure which diffracts green light with an efficiency of 78.6% and red and blue light with an efficiency of 69.7% [11]. Zhang and Chen et al. proposed a transparent near-eye display with an exit pupil expander (EPE) consisting of two multiplexed slanted gratings [12]. Jiasheng Xiao et al. designed a surface microstructure with a triple-carved-sub-grating, the period of which is 18.9 microns. It can be used in diffractive waveguide near-eye display devices [13]. HoloLens uses three layers of waveguides with slanted gratings to realize full-color display and two-dimensional exit pupil replication [14]. Li Shubin et al. proposed a tilted grating for optical communication [15]. Zhujun Shi et al. proposed a waveguide display design based on polarization-dependent grating. They use slanted gratings and rectangular gratings to extract TM light and TE light, respectively [16]. Muhammad Fayyaz Kashif et al. presented a two-dimensional dielectric grating; it can achieve high absorption in monolayer graphene in visible and near-infrared frequencies [17]. Jun Wu et al. proposed a pair of stacked metal-dielectric grating structures which are polarization-independent spectrum selective absorbers. The absorber can maintain a high absorption rate around a wide range of incidence angles (0–33°) [18]. However, about 60% of light is wasted in the light coupling when a symmetrical structure is used [19]. Therefore, in order to achieve high diffraction efficiency, it is crucial to design an asymmetric diffraction grating structure.

Most reported gratings achieving high diffraction efficiency at the −1 order under normal incidence are 1D gratings, or involve complex combinations of multiple 1D slanted gratings. For one-dimensional gratings, optical waveguides can achieve only one-dimensional pupil expansion in one direction, with low integration. In order to induce propagation for the optical waveguide in two directions, one method is to use a turning grating group, such that the optical path turns form two directions of pupil dilation. The other is to use two-dimensional diffraction grating. The advantage of two-dimensional grating is that there are only two grating areas, which reduce the loss of light in the propagation. In this work, we propose a highly efficient two-dimensional slanted grating for TE and TM polarization based on a double-layer cylindrical structure which is designed using the rigorous coupled wave analysis (RCWA) [20–23] and simulated annealing algorithms (SA) [24,25]. Our grating is a two-dimensional structure with high diffraction efficiency of the $(0, -1)$ and $(-1, 0)$ order. The slanted angle is 20.6°, which can greatly reduce the production difficulty. The $(0, -1)$ and $(-1, 0)$ order diffraction efficiencies of the grating exceed 35% for normal incidence in the range of 429–468 nm, and the total efficiency is up to 78%. In addition, we analyzed the manufacturing tolerance of the grating, which demonstrates its manufacturing feasibility and its application prospect in the AR/VR industry.

## 2. Design of the 2D Slanted Grating

According to the background mentioned above, two-dimensional slanted grating has more advantages than one-dimensional slanted grating. Considering the actual processing technology, we designed a two-dimensional slanted grating based on the cylindrical structure. In addition, a single-layer structure was difficult to obtain. However, in order to achieve high diffraction efficiency, we designed a multilayer structure to obtain higher diffraction efficiency and better performance.

Diffraction occurs when light incidents on the surface of a grating. Depending on the wavelength of incident light, angle of incidence, grating period, etc., different diffraction orders can be generated. The two-dimensional grating has periodicity in two different directions. The diffraction equation of a two-dimensional grating can be expressed as:

$$\sin\theta_{m,n}\cos\phi_{m,n} = \sin\theta\cos\phi + m\lambda/d_x \qquad (1)$$

$$\sin\theta_{m,n}\sin\phi_{m,n} = \sin\theta\sin\phi + n\lambda/d_y \qquad (2)$$

where m and n represent the diffraction order of the two-dimensional grating, and both m and n are integers. θ and φ are the polar and azimuthal angle of the incident light,

and $\theta(m,n)$ and $\phi(m,n)$ are the polar and azimuthal angle of the corresponding diffraction orders, respectively. $\lambda$ is the incident wavelength, and $d_x$ and $d_y$ are the periods of the grating along the x- and y-directions. Under normal incidence, the polar and azimuthal angles of the incident wave are both 0 degrees. We assume that both $d_x$ and $d_y$ are equal to d. Thus, the diffraction Equations (1) and (2) can be written as:

$$\sin^2 \theta_{m,n} = \left(m^2 + n^2\right)\lambda^2/d^2 \tag{3}$$

In order to achieve high diffraction efficiency (DE) for the $(-1, 0)$ and $(0, -1)$ diffraction orders, the total number of possible diffracted modes should be reduced as much as possible, so that the dispersion of the total energy by the unnecessary diffraction orders can be decreased. Therefore, the grating period d should satisfy the following condition:

$$\lambda < d < \sqrt{2}\lambda \tag{4}$$

The proposed 2D grating has a structure of a slanted two-layered nano-cylinder array, as shown in Figure 1a. The material of the bottom layer is $MgF_2$ with a height of $h_1$. The material of the top layer is $SiO_2$ with a height of $h_2$. The materials are chosen as their refractive indices are not sensitive to wavelength in the visible range, and their manufacture is cost effective. The refractive indices of $MgF_2$ and $SiO_2$ at the operating wavelength of 450 nm are 1.38 and 1.46, respectively. $MgF_2$ crystal is widely used, and $MgF_2$ is the transparent solid material with the smallest refractive index (n = 1.38). In addition, the extinction coefficient of $MgF_2$ is quite small in a large spectral range, which can decrease the reflection. Moreover, this material has a high laser-damage threshold [26–28]. The substrate material of the grating is $SiO_2$, which is the same as that of the top layer. The proposed 2D gratings are shown in Figure 1b–d, with the grating ridge width b, grating period d, grating polar angle $\alpha$, grating azimuthal angle $\beta$, and duty cycle *f*. To ensure the polarization-independent properties of the 2D grating, the periods are set to be equal in the x- and y-axis directions, $d = d_x = d_y$, and the grating azimuthal angle $\beta$ is 45 degrees. Duty cycle f is defined as the ratio of the ridge width to the period of the grating, which can be expressed as $f = b/d$, so $f = f_x = f_y$. The incident layer and grating slots above the grating structure are filled with air media.

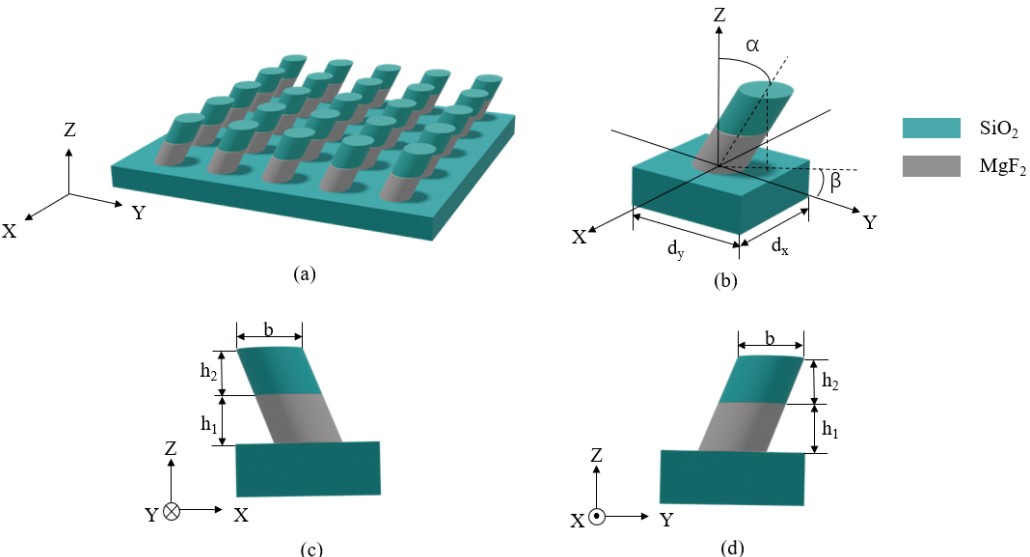

**Figure 1.** (**a**) The proposed 2D slanted grating structure; (**b**) unit-cell of the proposed structure; (**c**) unit-cell seen from side view (X–Z plane); (**d**) unit-cell seen from side view (Y–Z plane).

We use RCWA and SA algorithms for grating design and optimization of the parameters {d, f, $\alpha$, $h_1$, $h_2$}. The rigorous coupled-wave analysis (RCWA) is a purely numerical method. The rigorous coupled-wave approach analyzes the diffraction of an electromagnetic plane wave incident obliquely at a planar grating bounded by two different media. RCWA expands the electromagnetic field and the dielectric constant of the material by a Fourier series, and solves Maxwell's equations by solving the eigen-values and eigenvectors of the matrix. Therefore, RCWA is an effective tool for dealing with periodic structures, especially diffraction gratings. SA is a probabilistic technique that can approximate the global optimum of a given function. The purpose is to satisfy the requirements of higher efficiencies in the (0, −1) order and the (−1, 0) order, and lower efficiency in the (0, 0) order; thus, the cost function $\Phi$ was set up as follows:

$$\Phi = \frac{\left[DE_{(0,-1)} - DE_{av}\right]^2 + \left[DE_{(-1.0)} - DE_{av}\right]^z}{\sum_{i=-m,j=-n}^{i=m,j=n} DE_{(i,j)}} \tag{5}$$

$$I_{av} = \frac{1}{2}\left[DE_{(0,-1)} + DE_{(-1,0)}\right]$$

where $DE_{(0,-1)}$ and $DE_{(-1.0)}$ are the diffraction efficiencies of (0, −1) and (−1, 0) orders, and $DE_{(i,j)}$ is the diffraction efficiency of the (i,j)th order, respectively. DE is calculated by rigorous coupled-wave analysis. When $\Phi$ reaches the minimum value, the SA algorithm was cut off, and the optimal parameters solution were obtained.

The 2D grating structure parameters were processed with a global optimization, and the optimized parameters are d = 603 nm, $h_1$ = 330 nm, $h_2$ = 528 nm, f = 0.47, and $\alpha$ = 20.6° when the incident light at 450 nm is normally incident on the grating surface. With this structural parameter, the diffraction efficiencies of the (−1, 0) and (0, −1) orders are 39.10% and 39.12%, respectively, under the TE polarization incidence. When the incident light is TM polarized, the diffraction efficiencies of (−1, 0) and (0, −1) orders are 39.13% and 39.19%, respectively, at a wavelength of 450 nm.

According to the parameters, we can deduce that the polarization-dependent loss (PDL) of this grating is 0.01 db, which means a good polarization-independent characterization was achieved.

## 3. Characters and Tolerance Analysis

We obtained the parameters of the designed 2D slanted gratings. In order to achieve the performance of these designed gratings, we also analyzed the 2D slanted gratings by rigorous coupled wave analysis.

The wavelength bandwidth of a grating is a range of wavelengths over which the grating can operate effectively. A wider bandwidth means that the grating can be applied to a wider range of wavelengths, such that it is necessary to analyze the bandwidth of the grating for both polarizations. In Figure 2, we plot the diffraction efficiencies of the (−1, 0) and (0, −1) orders at a range of wavelengths from 400 nm to 500 nm for TE and TM polarization, respectively. Figure 2a shows the efficiencies of the diffraction orders (0, −1) and (−1, 0) for TE polarization incidence, and Figure 2b shows the efficiencies of (0, −1) and (−1, 0) for TM polarization incidence. We can see that the (−1, 0) order diffraction efficiency and (0, −1) order diffraction efficiency are both greater than 30% when the grating is operated in the range of 403 nm–490 nm wavelengths. In particular, the (−1, 0) order diffraction efficiency and the (0, −1) order diffraction efficiency are both higher than 35% at operating wavelengths of 429–468 nm. Hence, the 2D slanted gratings can be used in the wavelength of the red wave band.

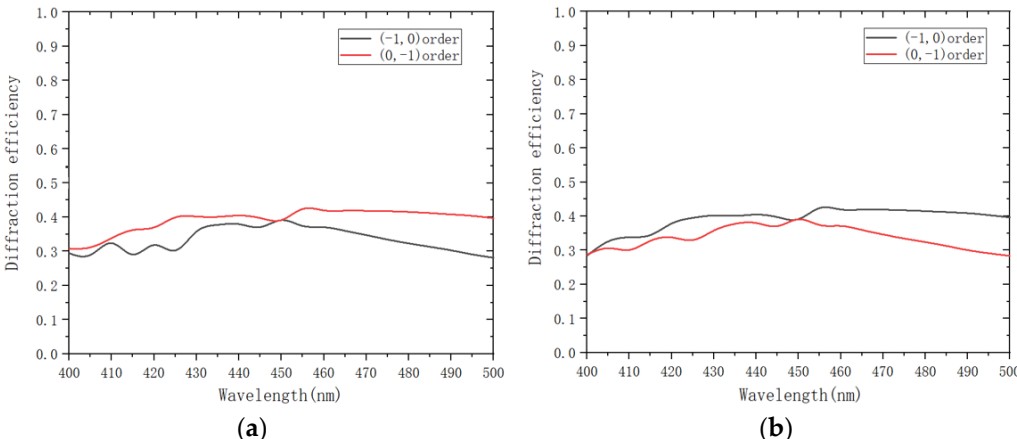

**Figure 2.** Grating diffraction efficiency versus incident wavelength (**a**) TE polarization; (**b**) TM polarization.

Figure 3 depicts the efficiency of (0, 0) versus wavelength under normal incidence for TE and TM polarization. We should decrease the efficiency of the 0th order to reduce the loss. The diffraction efficiencies of the (0, 0) order for TE and TM polarization are both less than 10% when the incident wavelength is in the range of 400 nm to 470 nm. In particular, the (0, 0) order diffraction efficiencies for TE and TM polarization are both less than 5% at operating wavelengths from 420 nm to 450 nm.

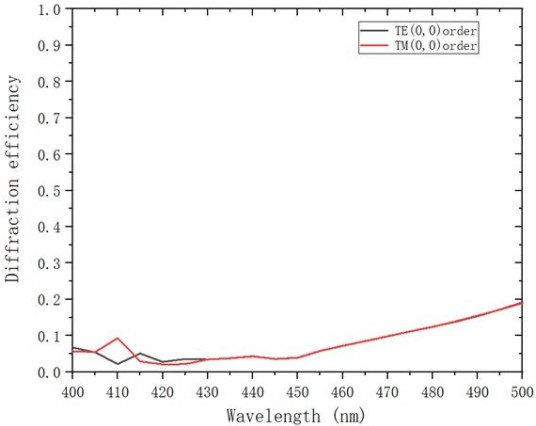

**Figure 3.** The efficiency of (0, 0) versus wavelength for TE and TM.

According to Figures 2 and 3, we can find that this 2D slanted grating has a better performance of beam splitting and a wide spectrum for TE and TM polarization, which demonstrates the ease of use.

In present times, these head-up display devices and near-eye display devices need a wider field of view (FOV). We also analyzed the efficiency versus incident angle because of the use of these gratings in the field of AR/VR technology. Figure 4 shows the influence of the incident angle on diffraction orders with optimized parameters. In the ideal case, when the angle of incidence is $0°$, the diffraction efficiency of $(-1, 0)$ and $(0, -1)$ are equal. However, when we increase the incident angle, their changes are opposite. The diffraction efficiency of $(-1, 0)$ is decreasing, whereas that of $(0, -1)$ is increasing. We can see that the diffraction efficiencies of the $(-1, 0)$ order and the $(0, -1)$ order are both greater than 30% when the incident angle ranges from $-3.2°$ to $3.2°$.

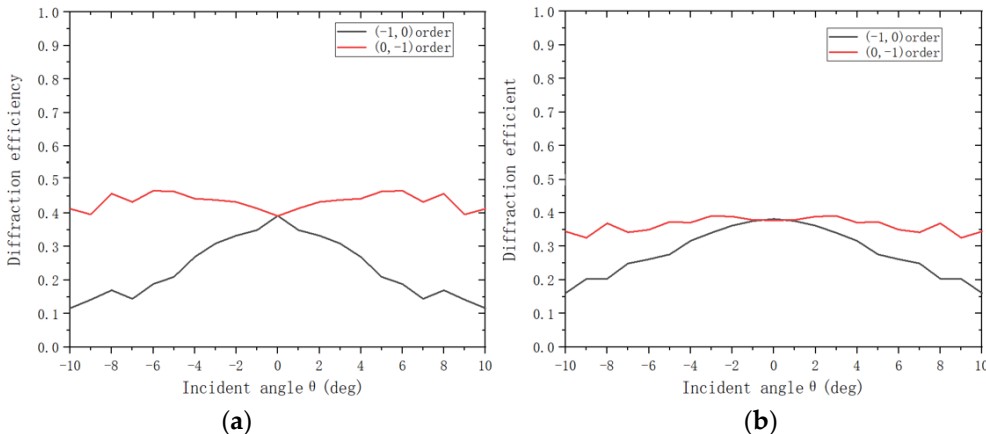

**Figure 4.** Diffraction efficiency versus incident angle for a wavelength of 450 nm with the optimized grating parameters. (**a**) Diffraction efficiency under TE polarization; (**b**) diffraction efficiency under TM polarization.

For general rectangular gratings, they are relatively easy to manufacture by using conventional lithography processes such as electron beam lithography, etching processes, and nano-imprint lithography. The ridge of traditional 2D gratings is perpendicular to the substrate. Tiziana Stomeo et al. fabricated a two-dimensional array by using an optimized nano-imprint lithography [29]. The production of slanted gratings has also been studied in a number of works throughout history. Li et al. produced a tilted grating with a tilt angle of 50° [30]. Tapani Levola et al. fabricated slanted gratings by using high refractive index materials and UV replication technology [31]. Figure 5 is the diagram of the fabrication process. In the fabrication, the $MgF_2$ layer is coated on the $SiO_2$ substrate at first. Then, a new layer of $SiO_2$ is coated on top of $MgF_2$. Thus, a triple-layer structure of $SiO_2$-$MgF_2$-$SiO_2$ is formed. Next, a layer of photoresist is coated on the surface evenly. The photoresist was exposed by using holographic or electron-beam direct-writing techniques, and the photoresist grating pattern is developed with a developer solution. Because the structure of the designed gratings is 2D slanted, we should perform exposure by lithography twice. The cylindrical structure can be formed after two rounds of exposure. In the real processing of fabrication, the slanted gratings could not be designed with a rectangular structure. Thus, we designed the slanted gratings with a cylindrical structure. Subsequently, we use a tilted incidence reactive ion beam for etching [32]. The 2D slanted grating will be fabricated after the etching process.

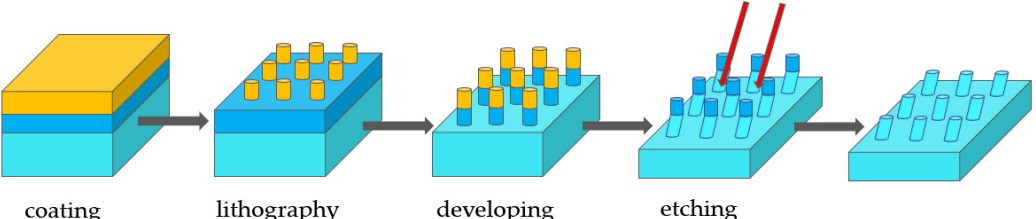

**Figure 5.** The diagram of fabrication processing.

During the experiments, the actual structure of the grating may slightly deviate from the optimized results. The process of coating can influence the depth of the film. The lithography can influence the duty cycle and period. Moreover, the process of etching can influence the slanted angle. Therefore, it is necessary to analyze the production tolerances for the grating parameters to obtain the appropriate range of grating structure data. This can ensure that the grating structure manufactured by experiment conforms with the designed parameters. The thickness of $MgF_2$ and $SiO_2$ is related to the phase modulation

of the incident light, which further affects the coupling of the efficiency of each diffraction order of the grating.

Figure 6 shows the effect of $MgF_2$ and $SiO_2$ thickness on the diffraction efficiency for the optimal solution of other structural parameters. Figure 6a,b show the $(-1, 0)$ order diffraction efficiency and the $(0, -1)$ order diffraction efficiency for the normal incidence of the TE polarized incident light. Figure 6c,d represents the condition of normal incidence with TM polarization. When the thickness $h_1$ of $MgF_2$ varies in the range of 300–350 nm and the thickness $h_2$ of $SiO_2$ varies in the range of 513–550 nm, the $(-1, 0)$ order diffraction efficiency and the $(0, -1)$ order diffraction efficiency both exceed 35% for TE and TM polarizations. The total effective efficiency of the grating is greater than 70%. In particular, the total effective efficiency is greater than 75% when the $MgF_2$ thickness is from 300 nm to 350 nm and the $SiO_2$ thickness is from 525 nm to 550 nm.

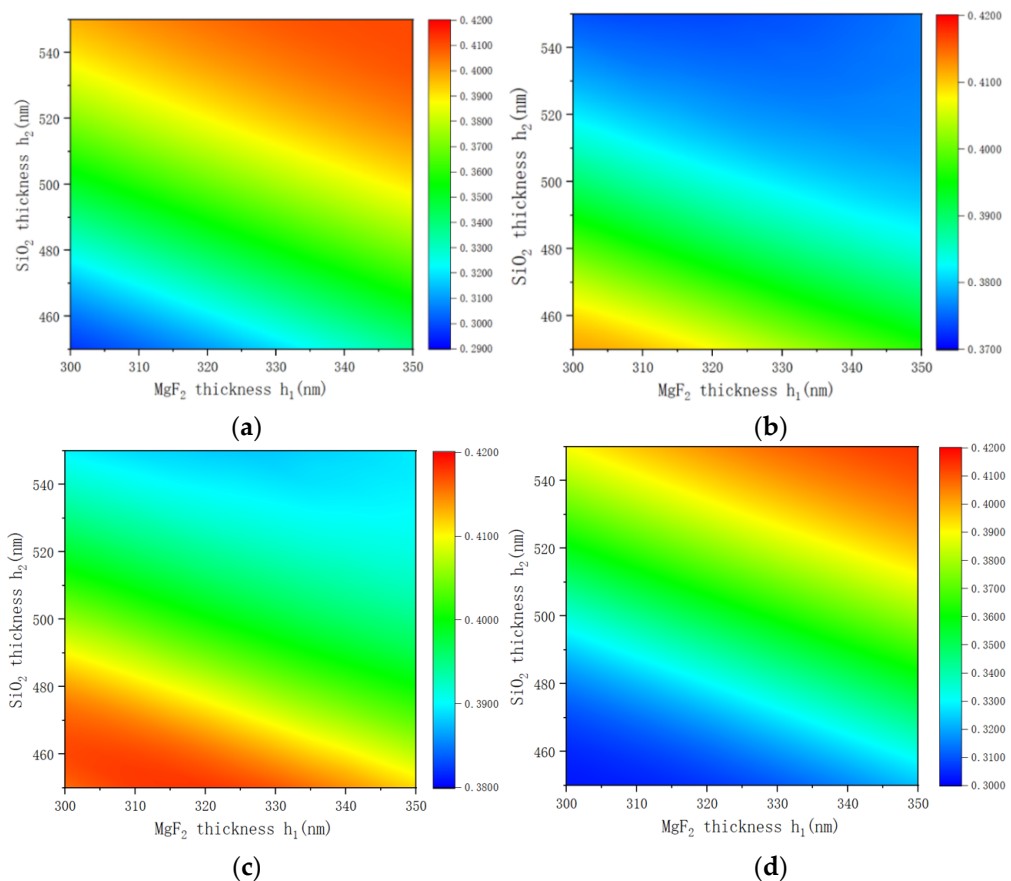

**Figure 6.** Grating diffraction efficiency versus thickness h1 of $MgF_2$ and thickness $h_2$ of $SiO_2$; (**a**) $(-1, 0)$ order diffraction efficiency under TE polarization; (**b**) $(0, -1)$ order diffraction efficiency under TE polarization; (**c**) $(-1, 0)$ order diffraction efficiency under TM polarization; (**d**) $(0, -1)$ order diffraction efficiency under TM polarization.

Figure 7 shows the $(-1, 0)$ and $(0, -1)$ order diffraction efficiency for different grating periods d and grating polar angle $\alpha$ with respect to the normal incidence for the optimal solution of the other structural parameters of the grating. Figure 7a,b are for TE polarized incidence, and Figure 7c,d are for TM polarized incidence. In the region of interest, for TE and TM polarization, the desired results are obtained when the grating period is varied from 580 nm to 633 nm and the grating slanted angle is varied from 12.2° to 21.0°. In this range, both the $(-1, 0)$ order diffraction efficiency and the $(0, -1)$ order diffraction efficiency all exceed 35%. During the grating production process, the control of the grating slanted angle is more demanding than the grating period. Our grating period has a 53 nm

fabrication tolerance, and the slanted angle has a 8.8-degree fabrication tolerance. This is a positive result. We have a relatively large range of manipulation in the manufacture process.

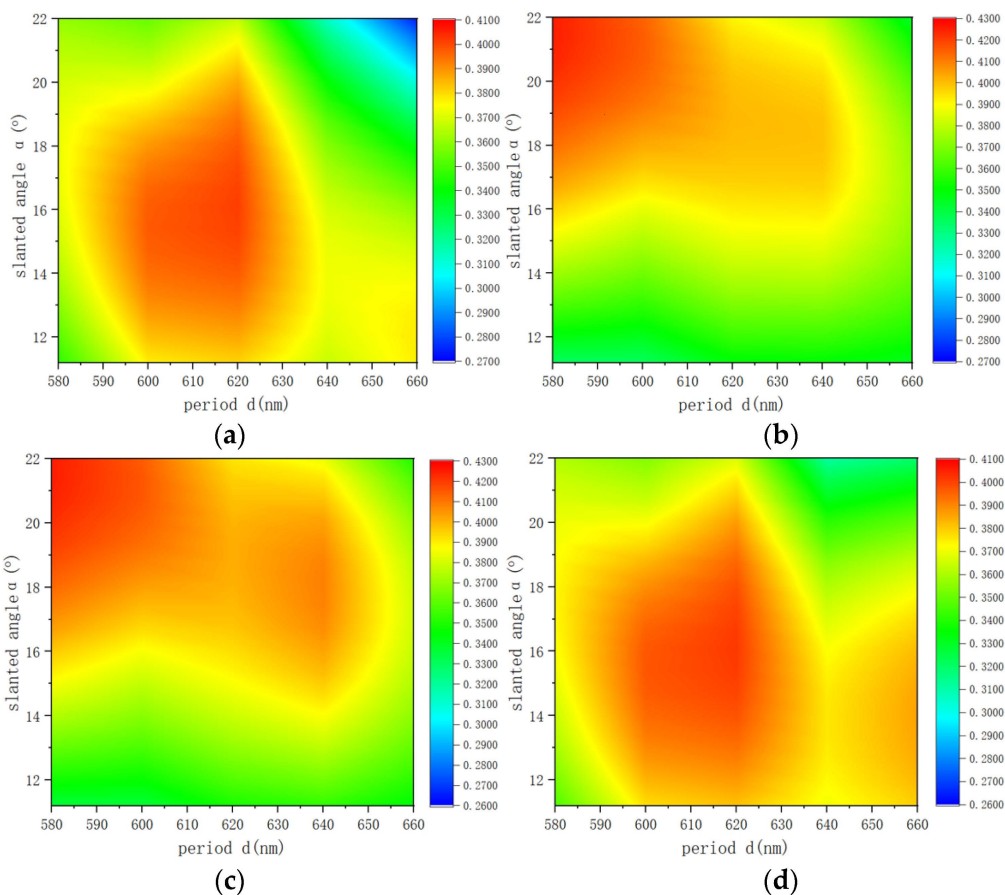

**Figure 7.** Grating diffraction efficiency versus grating period d and slanted angle $\alpha$; (**a**) $(-1, 0)$ order diffraction efficiency under TE polarization; (**b**) $(0, -1)$ order diffraction efficiency under TE polarization; (**c**) $(-1, 0)$ order diffraction efficiency under TM polarization; (**d**) $(0, -1)$ order diffraction efficiency under TM polarization.

Due to the characteristics of the existing fabrication technique, it is more difficult to control the duty cycle than the period or the height of the grating in the process of fabricating the grating. Therefore, it is necessary to calculate the production tolerance for the duty cycle and discuss the deviation of the duty cycle from the designed optimal values. In Figure 8, we show a tolerance calculation for the duty cycle, such that Figure 8a is for TE polarization incidence, and Figure 8b is for TM polarization incidence. In the region of interest, the $(-1, 0)$ order diffraction efficiency and the $(0, -1)$ order diffraction efficiency both exceed 35% when the duty cycle is in the range of 0.46 to 0.53. In this range, the grating ridge width range b is 277.38 nm to 319.59 nm when the grating period d is equal to the optimal value of 603 nm. The tolerance range of the ridge width of the grating is 42.21 nm. With the grating ridge width b in this range, the sum of the grating $(-1, 0)$ order diffraction efficiency and $(0, -1)$ order diffraction efficiency will be greater than 70%. Moreover, we can see that the diffraction efficiency of the $(-1, 0)$ and $(0, -1)$ orders decreases rapidly when the duty cycle is greater than 0.53, regardless of whether the incident wave is in the TE or TM polarization state.

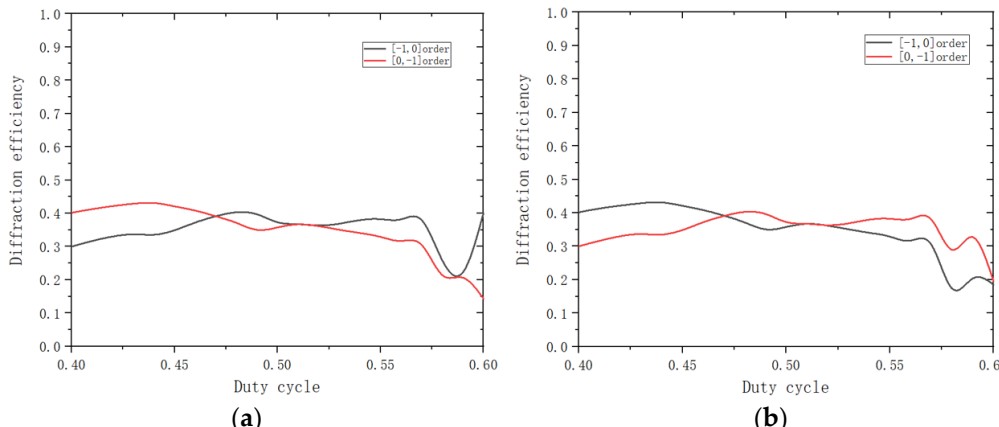

**Figure 8.** Grating diffraction efficiency versus duty cycle $f$; (**a**) TE polarization; (**b**) TM polarization.

According to our knowledge, this paper is the first to report a two-dimensional grating based on a double-layer cylindrical structure. The high diffraction efficiency and low PDL can improve the SNR. In consideration of the broadband for application and great tolerances for fabrication, the proposed 2D slanted gratings might be highly interesting for fabrication of high-quality 2D grating. Furthermore, the fabricated 2D slanted gratings might provide the highest efficiency with polarization independence, which implied the advanced revolutionary potential for AR/VR technology.

## 4. Conclusions

Diffraction gratings play an increasingly important role in various planar optical systems. At present, no one has proposed a 2D slanted grating suitable for 3D display. In this work, we propose a two-dimensional slanted grating based on a double-layer cylindrical structure by using rigorous coupled-wave analysis. The two-dimensional slanted grating was proposed based on a two-layer cylindrical structure, which can achieve high diffraction efficiencies for the $(0, -1)$ and $(-1, 0)$ orders under normal incidence for TE and TM polarization. The maximum total efficiency of the grating is up to 78%. The diffraction efficiencies of the $(-1, 0)$ order and the $(0, -1)$ order are both greater than 30% when the incident angle ranges from $-3.2°$ to $3.2°$. Tolerance analysis was performed for each structural parameter of the grating. Suitable manufacturing tolerances were calculated for the height of the two-layer cylindrical structure, grating inclination, and duty cycle, and the grating period deviated from the optimal results. The total effective efficiency is greater than 75% when the MgF$_2$ thickness is from 300 nm to 350 nm, and the SiO$_2$ thickness is from 525 nm to 550 nm. Furthermore, the grating period has a 53 nm fabrication tolerance, and the slanted angle has a 8.8-degree fabrication tolerance. The relatively large tolerances ensure that it is easy to fabricate the two-dimensional slanted grating and to achieve the targeted objectives. The proposed two-dimensional slanted grating can be applied to 2D exit pupil expansion, and the easy availability of materials greatly reduces the cost of production manufacturing, which has the potential to meet AR/VR and 3D display requirements.

**Author Contributions:** Y.C., J.W. and Y.W. conceived the idea of this article. Y.C. and X.L. completed the work of acquisition of data. G.J. and C.Z. shared the task of analysis and interpretation of data. J.W. and C.Z. provided funds for the project. All authors have read and agreed to the published version of the manuscript.

**Funding:** This research was funded by the Guangdong Provincial Pearl River Talents Program (2019ZT08Z779); the Guangzhou Science and Technology Program Key Projects (202007010001); the National Natural Science Foundation of China (U21A20509); and the National Natural Science Foundation of China (62205124).

**Institutional Review Board Statement:** Not applicable.

**Informed Consent Statement:** Not applicable.

**Data Availability Statement:** Not applicable.

**Conflicts of Interest:** The authors declare no conflict of interest.

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
