# Peer review of "A 1 × 2 Two-Dimensional Slanted Grating Based on Double-Layer Cylindrical Structure"

_applsci, doi:10.3390/app13042270_

Round 1

Reviewer 1 Report

The authors have presented a 2-dimensional double-layer dielectric grating. The structure has been numerically analyzed using the RCWA method and the geometry has been optimized using the SA optimization algorithm.  Overall I have found the paper interesting and well-written. However, I have the following suggestions/comments.

1. What does "1x2 two-dimensional" means in the title?

2. The introduction section can be improved specifically the first part where the applications of the diffraction gratings have been described. I would suggest adding a few more relevant applications/references such as optical absorbers for photodetection and sensing. Such as 

https://ieeexplore.ieee.org/abstract/document/6767108

https://www.mdpi.com/2076-3417/9/12/2528

https://link.springer.com/article/10.1007/s00339-019-3101-z 

3. By reading the paper, I could not understand why the grating was designed at 450 nm? and why not any other wavelength in VIS. This should be clarified. 

4. As mentioned while describing Fig.5, the total effective efficiency of the grating is 75%. It would be good for the reader to know what does total effective efficiency means. Does it mean the total efficiency of all refracted orders?

5. There is no mention of the material refractive index used in this study. It should be included. 

6. The designed grating structure is cylindrical. However, Fig.4 shows the fabrication process for rectangular grating. Why is it so? I would suggest including a few more relevant papers regarding the fabrication of pillar gratings. For example

[1]. https://www.mdpi.com/2076-3417/9/18/3812/htm  

7. The grating has been optimized and studied against all parameters except the incidence angle. It would be great to see the response for +-3 degrees of the incident angle.   

8. I would suggest modifying the following sentences. 

The production of slanted gratings has also been studied in a number of works throughout history.

At present, no one has proposed a 2D slanted grating suitable for 3D display.

Author Response

Reviewer #1:

The authors have presented a 2-dimensional double-layer dielectric grating. The structure has been numerically analyzed using the RCWA method and the geometry has been optimized using the SA optimization algorithm.  Overall I have found the paper interesting and well-written. However, I have the following suggestions/comments.

1.What does "1x2 two-dimensional" means in the title?

Response:

Thank you very much for your affirmation to our work first, then let me answer this question for you. “1×2” means that a light source is diffracted into two diffraction orders with similar efficiency after passing through the grating. The” two-dimensional” means that our grating structure is two-dimensional grating, our grating is periodic in two different directions.

Reviewer #1:

  1. The introduction section can be improved specifically the first part where the applications of the diffraction gratings have been described. I would suggest adding a few more relevant applications/references such as optical absorbers for photodetection and sensing. Such as

https://ieeexplore.ieee.org/abstract/document/6767108

https://www.mdpi.com/2076-3417/9/12/2528

https://link.springer.com/article/10.1007/s00339-019-3101-z

Response:

Thank you for your precious comments, we agree with the reviewer’s comment and have added a few more relevant references. The added statements:(page1 and page 2, lines 43-48).

Reviewer #1:

  1. By reading the paper, I could not understand why the grating was designed at 450 nm? and why not any other wavelength in VIS. This should be clarified.

Response:

Thank you for your question. Our 2D slanted gratings are designed to be used in AR/VR and 3D display, the grating needs to be suitable for visible wavelengths. But the 2D slanted gratings for visible wavelengths is quite hard to be designed, and we haven’t found designed parameters for the whole visible wavelengths. So we choose the 450 nm wavelength as incident light first. We are trying to design a 2D slanted grating which can be used for RGB wavelengths.

Reviewer #1:

  1. As mentioned while describing Fig.5, the total effective efficiency of the grating is 75%. It would be good for the reader to know what does total effective efficiency means. Does it mean the total efficiency of all refracted orders?

Response:

Thank you for your reminder. There could be many orders when grating is diffracting. The unnecessary diffraction orders will disperse the energy. Our grating can achieve high diffraction efficiencies for (0,-1) and (-1,0) orders, other orders are unnecessary. “the total effective efficiency of the grating is 75%” means that the efficiencies sum of (0,-1) and (-1,0) orders is 75%. It dose not mean the total efficiency of all diffraction orders.

Reviewer #1:

  1. There is no mention of the material refractive index used in this study. It should be included.

Response:

Thank you for your suggestion. This is a neglected problem, in the revised manuscript, we added the sentences to describe this.

Added statements: (page 2, lines 92-93).

Reviewer #1:

  1. The designed grating structure is cylindrical. However, Fig.4 shows the fabrication process for rectangular grating. Why is it so? I would suggest including a few more relevant papers regarding the fabrication of pillar gratings. For example

[1]. https://www.mdpi.com/2076-3417/9/18/3812/htm 

Response:

Thank you for your advice, in the revised manuscript, we have checked carefully and corrected it. We also added more relevant papers regarding the fabrication of pillar gratings.

Added statements: (page 5, lines 172-173 and fig. 5).

Reviewer #1:

  1. The grating has been optimized and studied against all parameters except the incidence angle.

It would be great to see the response for +-3 degrees of the incident angle.  

Response:

Thank you for your precious comments, we agree with the reviewer’s comment. We added the response of incident angle in the revised manuscript.

Added statements: (page 4, lines 158-163 and fig. 4).

Reviewer #1:

  1. I would suggest modifying the following sentences.

 The production of slanted gratings has also been studied in a number of works throughout history.

At present, no one has proposed a 2D slanted grating suitable for 3D display.

Response:

Thank you for your suggestion, we deleted this sentence in the revised manuscript.

Reviewer 2 Report

The manuscript "A 1×2 Two-dimensional slanted grating based on double-layer cylindrical structure" by Yuda Chen and co-authors presents investigates the influence of the geometric parameters of the grating on the diffraction efficiency of the (-1, 0) and (0, -1) order. An original design of a diffraction grating is proposed and optimized. The influence on the diffraction efficiency of the deviation of the grating parameters from the optimal ones is considered. This research work may be important to future application. However, according to the following comments, I can not suggest to publish this manuscript in the present form.

1. The overall quality of the text is low, there are numerous typos and duplicate sentences (lines 53-55 and 58-60; Refs. 13 and 14).

2. What values of the refractive indices of SiO2 and MgF2 thin films were used for the calculations?

3. The orientations of the axes in Figure 1(a) Figure 1(e) are not consistent with each other. Accordingly, the location and orientation of the slanted cylinders is not clear. For example: the base of the cylinders is an ellipse, therefore, it is not clear how the conditions for equality of periods dx=dy and duty cycles fx=fy were simultaneously achieved. In manuscript, the polar angle a of the orientation of the cylinder axis is introduced. For a better understanding of grating geometry, the value of azimuth angle of orientation of the cylinder axis should be indicated.

4. The normal light incidence on a diffraction grating is investigated. What do authors mean by TE and TM polarization of (-1, 0) and (0, -1) order?

5. Why the spectral range of 400-500 nm was chosen for the study?

Author Response

Reviewer #2:

The manuscript "A 1×2 Two-dimensional slanted grating based on double-layer cylindrical structure" by Yuda Chen and co-authors presents investigates the influence of the geometric parameters of the grating on the diffraction efficiency of the (-1, 0) and (0-1) order. An original design of a diffraction grating is proposed and optimized. The influence on the diffraction efficiency of the deviation of the grating parameters from the optimal ones is considered. This research work may be important to future application. However, according to the following comments, I can not suggest to publish this manuscript in the present form.

  1. The overall quality of the text is low, there are numerous typos and duplicate sentences (lines 53-55 and 58-60; Refs. 13 and 14).

Response:

Thank you for your precious correction, we agree with the reviewer’s comment. After examination, we found that refs. 13 and 14 are duplicate. So we deleted the refs. 13 of the old version. In the revised manuscript, We modified this part of the statement.

Reviewer #2:

  1. What values of the refractive indices of SiO2 and MgF2 thin films were used for the calculations?

Response:

Thank you for your reminder. This is a neglected problem, in the revised manuscript, we added the following sentences to describe this.

Added statements: (page 2, lines 92-93).

Reviewer #2:

  1. The orientations of the axes in Figure 1(a) – Figure 1(e) are not consistent with each other. Accordingly, the location and orientation of the slanted cylinders is not clear. For example: the base of the cylinders is an ellipse, therefore, it is not clear how the conditions for equality of periods dx=dy and duty cycles fx=fy were simultaneously achieved. In manuscript, the polar angle aof the orientation of the cylinder axis is introduced. For a better understanding of grating geometry, the value of azimuth angle of orientation of the cylinder axis should be indicated.

Response:

Thank you for your reminder. This is a good suggestion. 2D grating slanted has a complex structure.

In the revised manuscript, we mark the grating polar angle α and grating azimuthal angle β in figure 1. We also add the description of the slanted angle in the article.

Added statements: (page 2, lines 95).

Reviewer #2:

  1. The normal light incidence on a diffraction grating is investigated. What do authors mean by TE and TM polarization of (-1, 0) and (0, -1) order?

Response:

Thank you for your suggestion. TE and TM polarization of (-1, 0) and (0, -1) order mean the polarization of diffracted light. TE polarization is P polarization, it indicates that there is a magnetic field component but no electric field component in the propagation direction. TM polarization is S polarization, it indicates that there is an electric field component but no magnetic field component in the propagation direction.

Reviewer #2:

  1. Why the spectral range of 400-500 nm was chosen for the study?

Response:

Thank you for your question. Our 2D slanted gratings are designed to be used in AR/VR and 3D display. We designed the 2D slanted gratings at the wavelength of 450nm. So we choose the spectral range of 400-500 nm.

Reviewer 3 Report

The submitted manuscript is entitled “A 1×2 Two-dimensional slanted grating based on double-layer cylindrical structure”. The presented study fits well with the scope of the journal and seems interesting to the readership.

The authors present a 1x2 transmission two-dimensional (2D) slanted grating based on a double-layer cylindrical structure. The effects of the geometric parameters of the grating on the diffraction efficiency were investigated using coupled-wave analysis (RCWA). The authors discuss the tolerances for the grating parameters and application to 2D exit pupil expansion in AR/VR applications. 

The authors should pay attention to the space between values and units; this problem frequently appears for ‘nm’ (eg, line 119).

The authors conclude that the fabricated 2D slanted gratings might provide the highest efficiency with polarization independence, which is important for AR/VR technology. A short comment on potential improvements in AR/VR and 3D display technology should be added. The authors should discuss application prospects in the AR/VR industry in more depth and the resulting improvements.

Please include your affiliation.

Author Response

Reviewer #3:

The submitted manuscript is entitled “A 1×2 Two-dimensional slanted grating based on double-layer cylindrical structure”. The presented study fits well with the scope of the journal and seems interesting to the readership.

The authors present a 1x2 transmission two-dimensional (2D) slanted grating based on a double-layer cylindrical structure. The effects of the geometric parameters of the grating on the diffraction efficiency were investigated using coupled-wave analysis (RCWA). The authors discuss the tolerances for the grating parameters and application to 2D exit pupil expansion in AR/VR applications. 

  1. The authors should pay attention to the space between values and units; this problem frequently appears for ‘nm’ (eg, line 119).

Response:

Thank you for your precious correction. We agree with the reviewer’s comment. We corrected this mistake in the revised manuscript.

Reviewer #3:

  1. The authors conclude that the fabricated 2D slanted gratings might provide the highest efficiency with polarization independence, which is important for AR/VR technology. A short comment on potential improvements in AR/VR and 3D display technology should be added. The authors should discuss application prospects in the AR/VR industry in more depth and the resulting improvements.

Response:

Thank you for your precious correction. We add a short comment in the revised manuscript.

Added statements: (page 8, lines 276-278).

Reviewer #3:

  1. Please include your affiliation.

Response:

Thank you for your reminder. We added the affiliation in the revised manuscript.

Added statements: (page 1, lines 5)

Reviewer 4 Report

Dear Authors,

Thank you for your work. The article is very specific. The authors conducted a deep modeling experiment. Considering revising, I have a few comments:

1) line 24-26
The use of diffractive waveguides for near eye display devices was not first proposed by Tapani Levola. I guess you meant specifically waveguides with slanted gratins. First patents with diffractive waveguide are from the late 1980s. Please, check.

2) line 78-80

Can you please comment on the meaning of Eq 3., especially if the polar angle is 0 degrees.

3) line 93

What do you mean by «.. their refractive indices are not sensitive to wavelength in the visible range»?

TIR condition?

4) the period 603nm for the wavelength, 40 the diffraction angle is around 30 degrees. But in my estimation the TIR angle is 42 degrees. Have you checked waveguide condition?

5) Have you actually manufactured the gratings? Are Figures 5 and 6 derived from simulation or from the experiment?

6) line 211

Some of the comments looks quite subjectively, for example, «This is a positive result.» I recommend to avoid that kind of judgments in scientific paper even the result seems positive.

7) line 175 «The process of coating can influence the depth of film»
I have a question considering fabrication. In the etching process the height of the bottom layer should be h1+h2 due to the coating? How does this influence look?

Best regards

Author Response

Reviewer #4:

Dear Authors,

Thank you for your work. The article is very specific. The authors conducted a deep modeling experiment. Considering revising, I have a few comments:

  1. line 24-26
    The use of diffractive waveguides for near eye display devices was not first proposed by Tapani Levola. I guess you meant specifically waveguides with slanted gratins. First patents with diffractive waveguide are from the late 1980s. Please, check.

Response:

Thank you for your precious correction. We agree with the reviewer’s comment. And we deleted this sentence in the revised manuscript.

Reviewer #4:

  1. line 78-80

Can you please comment on the meaning of Eq 3., especially if the polar angle is 0 degrees.

Response:

Thank you for your reminder. The diffraction equation of a two-dimensional grating can be expressed as:

IF the polar angles of the incident wave is 0 degrees. The value of “” is 0. We assume that the dx=dy=d and sum the squares of Equation 1 and Equation 2. Equation 3 is obtained by us eventually.

The Equation 3 is actually the diffraction equation when the incident angle is 0 actually.

Reviewer #4:

  1. line 93

What do you mean by «.. their refractive indices are not sensitive to wavelength in the visible range»?

Response:

Thank you for your reminder. It means that the refractive indices are relatively stable in the visible range.

Reviewer #4:

  1. the period 603nm for the wavelength, 40 the diffraction angle is around 30 degrees. But in my estimation the TIR angle is 42 degrees. Have you checked waveguide condition?

Response:

Thank you for your reminder. The wavelength bandwidth is 403 nm-490 nm. The period 603nm for the wavelength 403nm the diffraction angle is around 42 degrees, which is close to TIR angle. With the wavelength increase, the diffraction will become larger. The waveguide condition is rational.

Reviewer #4:

  1. Have you actually manufactured the gratings? Are Figures 5 and 6 derived from simulation or from the experiment?

Response:

Thank you for your reminder. This article is based on the simulation results now. We are trying to fabricate this grating in the future.

Reviewer #4:

  1. line 211

Some of the comments looks quite subjectively, for example, «This is a positive result.» I recommend to avoid that kind of judgments in scientific paper even the result seems positive.

Response:

Thank you for your precious correction. We agree with the reviewer’s comment and delete this sentence in the revised manuscript.

Reviewer #4:

  1. line 175 «The process of coating can influence the depth of film? I have a question considering fabrication. In the etching process the height of the bottom layer should be h1+h2 due to the coating? How does this influence look?

Response:

Thank you for your suggestion. The fabrication of slanted grating is a complex process. We apply the suitable thickness of coating on the substrate. Thickness  of MgF2 and SiO2  decide the height of the grating ridge. The whole etching depth is h1+h2. Fig 6 shows the grating diffraction efficiency versus thickness h1 of MgF2 and thickness h2 of SiO2

Round 2

Reviewer 2 Report

4. The normal light incidence on a diffraction grating is investigated. What do authors mean by TE and TM polarization of (-1, 0) and (0, -1) order?

Response:

Thank you for your suggestion. TE and TM polarization of (-1, 0) and (0, -1) order mean the polarization of diffracted light. TE polarization is P polarization, it indicates that there is a magnetic field component but no electric field component in the propagation direction. TM polarization is S polarization, it indicates that there is an electric field component but no magnetic field component in the propagation direction.

Comments to Response: The normal light incidence on a diffraction grating is investigated. “At normal incidence, the definitions of S- and P-components break down, the cross product in Equation 8.4 goes to zero, and basis vector for the S-component ŝ becomes undefined. Any pair of orthogonal unit vectors on the surface can be chosen since ŝ and have become degenerate.[Russell A. Chipman, Wai-Sze Tiffany Lam, Garam Young, Polarized Light and Optical Systems, CRC Press, Taylor & Francis Group (Page 548)]. Therefore, it is not clear how the S- and P-polarized components are oriented. Data about the chosen orientation of the S- and P-components should be added to the manuscript.

Additional comments:

- Figure 4 presented the dependence of the diffraction efficiency on the polar angle of incident light θ, but the azimuthal angle of incident light φ at which the calculation was carried out is not specified.

- DEav and Iav from Equations 5 and 6 are not defined in the manuscript.

Author Response

Comments to Response:

1、The normal light incidence on a diffraction grating is investigated. “At normal incidence, the definitions of S- and P-components break down, the cross product in Equation 8.4 goes to zero, and basis vector for the S-component ŝ becomes undefined. Any pair of orthogonal unit vectors on the surface can be chosen since ŝ and ṕ have become degenerate.” [Russell A. Chipman, Wai-Sze Tiffany Lam, Garam Young, Polarized Light and Optical Systems, CRC Press, Taylor & Francis Group (Page 548)]. Therefore, it is not clear how the S- and P-polarized components are oriented. Data about the chosen orientation of the S- and P-components should be added to the manuscript.

Response:

Thank you for your reminder. The S- and P-components of (-1,0) and (0, -1) orders are opposite. In fact, this grating is used for ARVR, diffracted light is received by human eyes. The polarization state of diffraction order is not the focus.

Additional comments:

- Figure 4 presented the dependence of the diffraction efficiency on the polar angle of incident light θ, but the azimuthal angle of incident light φ at which the calculation was carried out is not specified.

Response:

Thank you for your reminder. This grating is used for ARVR, the light is perpendicular to the human eye when the human looks straight ahead. We do not specify the calculation of the azimuthal angle φ of the incident light.

- DEav and Iav from Equations 5 and 6 are not defined in the manuscript.

 Response:

Thank you for your reminder. This is a neglected problem, in the revised manuscript, we added the  sentences to describe this.